# Theoretical investigations on Kerr and Faraday rotations in topological multi-Weyl Semimetals

**Supriyo Ghosh[1][⋆][∘], Ambaresh Sahoo[2][†][∘] and Snehasish Nandy[1,3,4][‡]**

**1** Department of Physics, University of Virginia, Charlottesville, VA 22904 USA
**2** Department of Physical and Chemical Sciences, University of L'Aquila,
Via Vetoio, L'Aquila 67100, Italy
**3** Theoretical Division, Los Alamos National Laboratory, Los Alamos, New Mexico 87545, USA
**4** Center for Nonlinear Studies, Los Alamos National Laboratory,
Los Alamos, New Mexico 87545, USA

⋆ sg4at@virginia.edu , † ambaresh.sahoo@univaq.it , ‡ snehasish12@lanl.gov

## Abstract

Motivated by the recent proposal of giant Kerr rotation in WSMs, we investigate the Kerr and Faraday rotations in time-reversal broken multi-Weyl semimetals (mWSMs) in the absence of an external magnetic field. Using the framework of Kubo response theory, we find that both the longitudinal and transverse components of the optical conductivity in mWSMs are modified by the topological charge ($n$). Engendered by the optical Hall conductivity, we show in the thin film limit that, while the giant Kerr rotation and corresponding ellipticity are independent of $n$, the Faraday rotation and its ellipticity angle scale as $n$ and $n^2$, respectively. In contrast, the polarization rotation in semi-infinite mWSMs is dominated by the axion field showing $n$ dependence. In particular, the magnitude of Kerr (Faraday) angle decreases (increases) with increasing $n$ in Faraday geometry, whereas in Voigt geometry, it depicts different $n$-dependencies in different frequency regimes. The obtained results on the behavior of polarization rotations in mWSMs could be used in experiments as a probe to distinguish single, double, and triple WSMs, as well as discriminate the surfaces of mWSMs with and without hosting Fermi arcs.

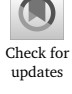
## Contents

---

∘ These authors contributed equally to the development of this work.

# 1 Introduction

The three-dimensional (3D) Dirac and Weyl semimetals have attracted tremendous attention to both theorists and experimentalists due to their unique band topology of late. The Weyl semimetals (WSMs) appear as topologically nontrivial conductors containing gapless chiral quasiparticles, known as Weyl fermions, near the touching of a pair of non-degenerate bands, also called "Weyl nodes" [1–8]. The nontrivial topological properties of WSMs are encapsulated by the Weyl nodes, which can act as a source or sink of the Abelian Berry curvature, an analog of the magnetic field but defined in the momentum space with quantized Berry flux. Each Weyl node is associated with a chirality quantum number, known as the topological charge, whose strength is related to the Chern number and is quantized in integer values [9]. According to the no-go theorem, in the case of a WSM, the Weyl nodes always come in pairs of positive and negative monopole charges and the total monopole charge summed over all the Weyl nodes in the Brillouin zone vanishes [10, 11].

Recently the materials such as TaAs, $MoTe_2$, $WTe_2$ etc., where WSM-phase has been realized experimentally, the topological charge ($n$) associated with the Weyl nodes are equal to $\pm 1$ [12–17]. Interestingly, WSMs containing Weyl nodes with higher topological charge $n > 1$, namely, the multi-Weyl semimetals (mWSMs), have been proposed to realize in condensed matter systems [7, 18–20]. Compared to the single WSM, whose energy dispersion is linear along all momentum directions (i.e., isotropic dispersion), the mWSMs ($n > 1$) show natural anisotropy in dispersion. In particular, both the double WSM ($n = 2$) and triple WSM ($n = 3$) depict linear dispersion along one symmetry direction; however, they exhibit quadratic and cubic energy dispersion, respectively, for the other two directions. Using the density functional theory (DFT) calculations, it has also been proposed that the double WSM phase can be realized in $HgCr_2Se_4$ and $SrSi_2$ [7, 18, 19], whereas $A(MoX)_3$ (with A = Rb, TI; X = Te) can accommodate triple-Weyl points [21]. It is to be noted that, the topological charge associated with Weyl nodes in real materials cannot be greater than 3 ($n \leq 3$) due to restriction arising from discrete rotational symmetry on a lattice [18, 20]. Moreover, the single WSM can be viewed as a 3D-analog of graphene, whereas the double WSM and triple WSM can be represented as 3D counterparts of bilayer and ABC-stacked trilayer graphene, respectively [22–24].

Topological semimetals exhibit a plethora of intriguing optical phenomena due to their unique band topology in the presence as well as in absence of external fields [17, 25–37]. The electrodynamic response of a WSM with broken time-reversal symmetry (TRS) has been a prime topic of interest to both theorists and experimentalists due to its connection to the axion field, which modifies the Maxwell's equations [38–40]. It has been shown in recent studies that giant Kerr rotation [41, 42], magneto-optical Kerr effect [43, 44], tunable perfect absorption [45] can occur in a single WSM. On the other hand, the electrodynamic response,

particularly the Kerr and Faraday rotations in the presence of higher monopole charge $n$, i.e., in mWSMs, has not been explored yet.

In this work, we study the Kerr and Faraday rotations ($\Phi_{K/F}$) in type-I TRS broken mWSMs in the absence of an external magnetic field to explore the effect of higher monopole charge ($n > 1$) on the polarization rotation. Using the Kubo response formula, we first analytically obtain the optical conductivity tensor for mWSMs. Our analytical expressions show that both the longitudinal and transverse components of the conductivity tensor are modified by the topological charge $n$. Specifically, the longitudinal components ($\sigma_{xx}, \sigma_{yy}$) perpendicular to the Weyl node separation ($Q\hat{z}$) as well as the transverse component ($\sigma_{xy}$) are proportional to $n$, whereas the other longitudinal component ($\sigma_{zz}$), which is along the node separation, shows nonlinear dependence on $n$. Using the obtained optical conductivity, we then study polarization rotation for two cases: (i) thin film limit of mWSMs and (ii) semi-infinite mWSMs. In the thin film limit of mWSMs, we show the Kerr rotation and corresponding ellipticity angle are independent of $n$ and vanish in the Pauli blocked regime. In contrast, the Faraday rotation and ellipticity angle of the transmitted light survive even in the Pauli blocked regime. Interestingly, we find that they are dependent on the topological charge and scale as $n$ and $n^2$, respectively. In addition, the polarization rotation angle turns out to be very large compared to other materials.

In the case of semi-infinite mWSM, we investigate the polarization rotation for two configurations: (i) Faraday geometry and (ii) Voigt Geometry. Our analysis demonstrates that, unlike the case of thin film mWSMs, the axion electrodynamics comes into play and dominates (which modifies Maxwell's equations), giving rise to finite polarization rotations in both cases, even in the Pauli-blocked regime. We show that polarization rotation is a linear (quadratic) function of $Q$ in Faraday (Voigt) geometry. The magnitude of the Kerr (Faraday) angle decreases (increases) with increasing $n$ in Faraday geometry, whereas in Voigt geometry, it has different $n$-dependencies in different frequency regimes. Furthermore, we identify that in the Faraday (Voigt) geometry, circular (linear) birefringence and circular (linear) dichroism increase with $n$.

## 2 Model Hamiltonian

The low-energy effective Hamiltonian describing a Weyl node with topological charge $n$ and chirality $s$ can be written as [46–49]

$$H_n^s(\mathbf{k}) = s\hbar \left\{ \alpha_n k_\perp^n \left[ \cos(n\phi_k)\sigma_x + \sin(n\phi_k)\sigma_y \right] + \nu k_z^s \sigma_z \right\} + C_s \hbar \nu k_z^s - sQ_0, \tag{1}$$

where $k_\perp = \sqrt{k_x^2 + k_y^2}$, $k_z^s = (k_z - sQ)$, $\phi_k = \tan^{-1}(k_y/k_x)$, and $\sigma_i$'s $(\sigma_x, \sigma_y, \sigma_z)$ denote the Pauli matrices representing the pseudo-spin indices. The Weyl nodes of opposite chirality are shifted both in momentum space and energy space by $2Q$ (along the $z$-direction) and $\pm Q_0$ due to broken TRS and inversion symmetry (IS), respectively. Here, $\alpha_n = \nu_\perp / k_0^{n-1}$ with $\nu_\perp$ being the effective velocity of the quasiparticles in the plane perpendicular to the $z$ axis and $k_0$ represents a material-dependent parameter having the dimension of momentum. Also, we consider both the velocity ($\nu$) and tilt parameter ($C_s$) along the $z$-direction. Note that, in this work we restrict ourselves to a type-I multi-Weyl node, i.e., $|C_s| < 1$, which indicates that the Fermi surface is point-like at the Weyl node. The energy dispersion of the multi-Weyl node associated with chirality $s$ is given by $\epsilon_{\mathbf{k},s}^\pm = C_s \hbar \nu (k_z - sQ) - \hbar s Q_0 \pm \hbar \sqrt{\alpha_n^2 k_\perp^{2n} + \nu^2 (k_z - sQ)^2}$, where $\pm$ represent conduction and valence bands, respectively. It is now clear that for $\nu = \nu_\perp$, the dispersion around a Weyl node with $n = 1$ is isotropic in all momentum directions. On the other hand, for $n > 1$, we find that the dispersion around a double (triple) Weyl node becomes quadratic (cubic) along both $k_x$ and $k_y$ directions whereas varies linearly with $k_z$.

# 3 Optical conductivity in mWSMs

In this section, we analytically derive the optical conductivity tensor to investigate the Kerr and Faraday rotations in mWSMs. In doing so, we first review the light field induced interband optical transition probabilities. It is well known that the light field can couple to the electron of a system via i) orbital coupling (momentum $\mathbf{k}$ is replaced by $\mathbf{k} - e\mathbf{A}$ due to minimal coupling, with $\mathbf{A}$ being vector potential due to the light field) and ii) Zeeman coupling. Therefore, the total Hamiltonian of light-electron interaction can be written as

$$H_{le} = -\frac{e}{m_e c}\mathbf{v_k}.\mathbf{A} + g\mu_B(\nabla \times \mathbf{A})\cdot\sigma\,, \tag{2}$$

where $e$ is the charge of electron, $m_e$ is the mass of the electron, $g$ is the Lande-$g$ factor, $\mu_B$ is the Bohr magneton and $\mathbf{v_k} = \frac{1}{\hbar}\partial\mathcal{H}/\partial\mathbf{k}$ with $\mathcal{H}$ representing the non-interacting Hamiltonian of the considered system. Here, the first term of the above equation gives rise to orbital coupling while the second term ($\propto \nabla \times \mathbf{A}$) leads to Zeeman coupling. Also, the term associated with orbital coupling gives rise to the Berry-curvature-dependent interband transition. Moreover, we would like to point out that in this work, we neglect the Zeeman coupling with the light field.

In view of this, the interband optical conductivity $[\sigma_{mn}(\omega)]$ for multi-Weyl semimetal in the linear response regime using the Kubo formula can be written as

$$\sigma_{mn}^s(\omega) = -\lim_{\gamma\to 0} ie^2 \int [d\boldsymbol{k}]\frac{f_{\boldsymbol{k}}^{eq}}{\epsilon_{\mathbf{k},s}^+ - \epsilon_{\mathbf{k},s}^-}\sum_{\alpha,\beta,\alpha\neq\beta}\frac{\mathcal{P}_m^{\alpha\beta}(\boldsymbol{k})\otimes\mathcal{P}_n^{\beta\alpha}(\boldsymbol{k})}{\omega + \frac{1}{\hbar}\left(\epsilon_{\mathbf{k},s}^\alpha - \epsilon_{\mathbf{k},s}^\beta\right) + i\gamma}\,, \tag{3}$$

where $[d\boldsymbol{k}] = d^3\boldsymbol{k}/(2\pi)^3$, $\omega$ is the optical frequency, and $\gamma$ represents the phenomenological damping term for the interband coherence. Here, $f_{\boldsymbol{k}}^{eq} = f(\epsilon_{\mathbf{k},s}^+, \mu) - f(\epsilon_{\mathbf{k},s}^-, \mu)$ is the equilibrium population difference between the conduction band and the valance band with $\mu$ being the chemical potential. Now the optical transition matrix element $[\mathcal{P}^{\alpha\beta}(\boldsymbol{k})]$ ($\alpha$ and $\beta$ denote the band indices), which gives rise to vertical transition between valence and conduction bands, can be written as $\mathcal{P}^{-+}(\boldsymbol{k}) = \langle\psi^-|\boldsymbol{v_k}|\psi^+\rangle$, where $\psi^-$ and $\psi^+$ are respectively the Bloch wavefunctions of valence and conduction bands. The factor $\mathcal{P}_m^{\alpha\beta}(\boldsymbol{k})\otimes\mathcal{P}_n^{\beta\alpha}(\boldsymbol{k})$ is related to the Berry curvature $[\Omega_{mn}(\boldsymbol{k})]$ of the mWSM, where $m, n = x, y, z$ and $\otimes$ represents the outer product of the optical matrix elements.

Using the wavefunctions of the conduction and valence bands, the different components of the optical matrix element for the multi-Weyl Hamiltonian given in Eq. (1) can be obtained as

$$\mathcal{P}^{s,-+} = \begin{pmatrix} s\Gamma_n\cos\phi_k - i\Lambda_n\sin\phi_k \\[2mm] s\Gamma_n\sin\phi_k + i\Lambda_n\cos\phi_k \\[2mm] -s\Gamma_n\frac{k_\perp}{nk_z^s} \end{pmatrix}\,, \tag{4}$$

where $\Gamma_n = n\alpha_n k_\perp^{n-1}k_z^s v/\sqrt{\alpha_n^2 k_\perp^{2n} + v^2(k_z^s)^2}$ and $\Lambda_n = n\alpha_n k_\perp^{n-1}$. One can notice that the optical matrix is independent of tilt velocity $C_s$ and linearly proportional to $n$. Substituting these optical matrix elements into Eq. (3), we can calculate both the diagonal and off-diagonal components of the optical conductivity tensor $\sigma_{mn}^s(\omega)$. Considering the optical conductivity as $\sigma_{mn} = \sigma'_{mn} + i\sigma''_{mn}$, where $\sigma'_{mn}$ and $\sigma''_{mn}$ are the real and imaginary part of it, we first calculate the diagonal components of conductivity tensor in the following subsection.

We would like to emphasize that, in this paper, we consider a TRS broken mWSM containing two multi-Weyl nodes with opposite chirality ($s$) separated in the momentum space by $2Q$.

This is due to the fact that the Hall conductivity ($\sigma_{xy}$) in the linear response regime vanishes in the TRS invariant system. As a result, the Kerr and Faraday rotations, which are linearly proportional to $\sigma_{xy}$, vanish. Furthermore, we assume two different tilt configurations of the Weyl nodes: i) chiral-tilt (i.e., $C_+ = -C_-$) and ii) achiral-tilt (i.e., $C_+ = C_-$).

### 3.1 Diagonal components of optical conductivity

The components of the optical conductivity tensor of tilted mWSMs described by the model Hamiltonian [Eq. (1)] are calculated assuming the zero-temperature regime, where Heaviside step functions replace the corresponding Fermi-Dirac distribution functions. We calculate the real part of the diagonal or longitudinal components of the optical conductivity tensor for mWSM systems by evaluating the integral equation [Eq. (3)] with the help of principal value ($P$) equation of the Dirac identity: $\lim_{\gamma \to 0} 1/(z + i\gamma) = P \int_{-\infty}^{\infty} (1/z) - i\pi\delta(z)$, which takes the following form:

$$\text{Re}[\sigma_{mm}] = -\frac{1}{(2\pi)^3} \int_0^{2\pi} d\phi_k \int_0^\infty k_\perp dk_\perp \int_{-k_c}^{k_c} dk_z \frac{f_k^{eq}}{\hbar\sqrt{\alpha_n^2 k_\perp^{2n} + v^2 k_z^2}} |(\mathcal{P}^{-+})_m|^2 \pi\delta(\omega - \omega_k). \quad (5)$$

Note that, to avoid the principal term $P$ of the Dirac identity, we first calculate the $\text{Re}[\sigma_{mm}]$, and the $\text{Im}[\sigma_{mm}]$ is subsequently calculated using Kramers-Kronig relation. The chirality index $s$ in Eq. (5) is omitted by calculating the optical conductivity for the $s = 1$ node [this leads to $k_z^s \to k_z$ and $C_{s=1} = C$ (say)], which is identical for both nodes and thus multiplied by a factor of 2 that cancels with the factor 2 appearing in the denominator due to energy difference. In addition, $f_k^{eq} = \Theta(\mu - \hbar\epsilon_{\mathbf{k}}^+) - \Theta(\mu - \hbar\epsilon_{\mathbf{k}}^-)$, $\hbar\omega_k = \epsilon_k^+ - \epsilon_k^-$, and we set $Q_0 = 0$ for simplicity. The term $(\mathcal{P}^{-+})_m$ denotes the $m$-th component of the optical matrix element given in Eq. (4) and is the only term that depends on $\phi_k$.

To calculate the $x$-component of the conductivity $\text{Re}[\sigma_{xx}]$, we provide a very brief description of the process by which we arrive at the closed-form expressions. The procedure is the same for $y$ and $z$ components, and due to rotational symmetry in the $x - y$ plane, we will end up having $\sigma_{xx} = \sigma_{yy}$. To begin, we evaluate the $\phi_k$ integral with $|(\mathcal{P}^{-+})_m|^2$ term from Eq. (4). Then, with the change of variables: **(i)** $k_\perp = k_\perp \alpha_n^{-1/n}$, $k_z = k_z/v$; **(ii)** $k_\perp = k_\perp^{1/n}$, and with $\hbar\omega_k = 2\hbar k$, the integral equation transforms as:

$$\text{Re}[\sigma_{xx}(\omega)] = -\frac{e^2 n}{8\pi\hbar v} \int_0^\infty k_\perp dk_\perp \int_{-k_c/v}^{k_c/v} dk_z \frac{\Theta_- - \Theta_+}{k} \left[\frac{k_z^2}{k^2} + 1\right] \delta(\omega - 2k), \quad (6)$$

where $\Theta_\pm = \Theta[\mu - \hbar|C|k_z \pm \hbar k]$. Now, evaluating the $k$ integration with $\delta$-function for a fixed $k_z$ (i.e., $k^2 = k_\perp^2 + k_z^2 \to k\,dk = k_\perp dk_\perp$), and thereafter substituting $k_z = vk_z$, the $\text{Re}[\sigma_{xx}]$ can be expressed as

$$\text{Re}[\sigma_{xx}(\omega)] = \mathcal{G}_{xx} \int_{-1}^1 dx \left[1 + x^2\right] \left(\Theta_{x+} - \Theta_{x-}\right), \quad (7)$$

where $\mathcal{G}_{xx} = e^2 n \omega/(32\pi\hbar v)$ and $\Theta_{x\pm} = \Theta\left[(2\mu/\hbar\omega|C|) \pm (1/|C|) - x\right]$ with $x = 2vk_z/\omega$. In a similar way, one can calculate the $\text{Re}[\sigma_{zz}]$ component as

$$\text{Re}[\sigma_{zz}(\omega)] = \mathcal{G}_{zz} \int_{-1}^1 dx \left[1 - x^2\right]^{1/n} \left(\Theta_{x+} - \Theta_{x-}\right), \quad (8)$$

where $\mathcal{G}_{zz} = e^2 v \omega \alpha_n^{-2/n} (\omega/2)^{2(1-n)/n}/(16\pi\hbar n)$.

After evaluating the above integrals [Eqs. (7)-(8)], the real parts of the diagonal components of the conductivity of the type-I WSM for different frequency regimes are calculated as

$$\text{Re}[\sigma_{xx}(\omega)] = \begin{cases} 0, & \text{for } 0 < \omega < \omega_1, \\ \sigma_\omega^n \left(\frac{1}{2} - \kappa_d\right), & \text{for } \omega_1 < \omega < \omega_2, \\ \sigma_\omega^n, & \text{for } \omega > \omega_2, \end{cases} \tag{9}$$

where $\kappa_d = \left(\frac{2\mu}{\hbar\omega} - 1\right)\left[\frac{3}{8|C|} + \frac{1}{8|C|^3}\left(\frac{2\mu}{\hbar\omega} - 1\right)^2\right]$, $\sigma_\omega^n = e^2 n \, \omega/(12\pi\hbar v)$, and $\hbar\omega_{1,2} = 2\mu/(1\pm|C|)$ are the two photon energy bounds in the mWSM. It is clear from the Eq. (9) that in the region $\omega < \omega_1$, vertical transition is completely Pauli blocked ($\text{Re}[\sigma_{xx}] = 0$), while in the intermediate region $\omega_1 < \omega < \omega_2$ and the region toward right $\omega > \omega_2$, vertical transition is partially Pauli blocked and completely unblocked, respectively. In addition, when compared to single Weyl case ($n = 1$) [41], the $\text{Re}[\sigma_{xx}]$ in mWSM increases linearly with $n$. In the limit $C \to 0$, we have $\hbar\omega_1 \to \hbar\omega_2 \to 2\mu$. As a result, the range of Pauli blocked region broadens and the intermediate region disappears. In this case, $\text{Re}[\sigma_{xx}(\omega)]$ becomes finite for $\hbar\omega > 2\mu$ only. In contrast, when $C \to 1$, we have $\hbar\omega_1 \to \mu$ and $\hbar\omega_2 \to \infty$, implying that the intermediate region extends to a very high energy.

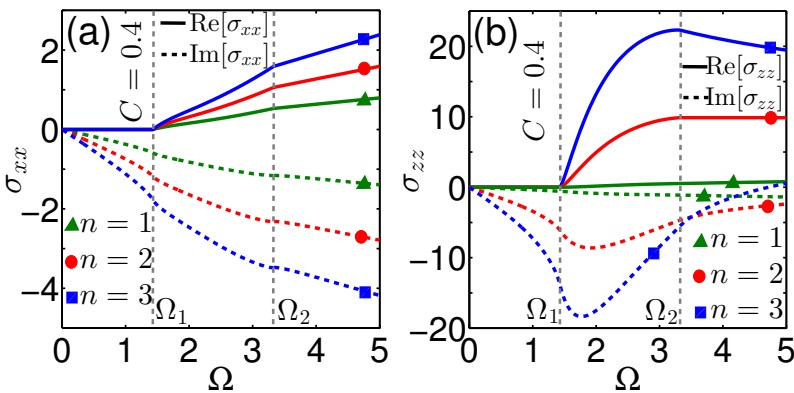

Figure 1: (a) Real (solid curves) and imaginary (dashed curves) parts of the $\sigma_{xx}$ as a function of rescaled frequency $\Omega \, (= \hbar\omega/\mu)$ of type-I ($C = 0.4$) mWSMs ($n = 1, 2, 3$). (b) Represents the same for $\sigma_{zz}$. Here, the vertical dashed lines represent the two photon frequency bounds $\Omega_{1,2} = \hbar\omega_{1,2}/\mu$. The conductivities are rescaled by $e^2 Q/(\pi h)$, and the values of other parameters are taken as: $Q \approx 5 \times 10^8 m^{-1}$, $v = 10^6 m/s$, $\mu = 0.1eV$, $k_0 = 0.8^{-1}$, $\alpha_2 = 1.25 \times 10^{-4} m^2/s$, $\alpha_3 = 1.56 \times 10^{-14} m^3/s$, and $\hbar\omega_c/\mu = 70$.

The imaginary part of the $\sigma_{xx}$ is followed from the Kramers-Kronig relation and takes the form:

$$\begin{aligned} \text{Im}[\sigma_{xx}(\omega)] = -\frac{\sigma_\omega^n}{4\pi} \Bigg\{ & \xi \ln\left|\frac{(\omega_2^2 - \omega^2)}{(\omega_1^2 - \omega^2)}\right| + \frac{8}{|C|^2}\left(\frac{\mu}{\hbar\omega}\right)^2 \\ & - \left(\frac{\mu}{\hbar\omega}\right)^3 \zeta(|C|, \omega, \mu)\ln\left|\frac{(\omega_2 - \omega)(\omega_1 + \omega)}{(\omega_1 - \omega)(\omega_2 + \omega)}\right| \\ & + \frac{6}{|C|^3}\left(\frac{\mu}{\hbar\omega}\right)^2 \ln\left|\frac{(\omega_2^2 - \omega^2)\omega_1^2}{(\omega_1^2 - \omega^2)\omega_2^2}\right| + 4\ln\left|\frac{\omega_c^2 - \omega^2}{\omega_2^2 - \omega^2}\right| \Bigg\}, \end{aligned} \tag{10}$$

where $\xi = (2 + \frac{3}{2|C|} + \frac{1}{2|C|^3})$, $\zeta(|C|, \omega, \mu) = \frac{4}{|C|^3} + 3\left(\frac{\hbar\omega}{\mu}\right)^2\left(\frac{1}{|C|^3} + \frac{1}{|C|}\right)$, and $\omega_c = vk_c$ is the cut-off frequency with $k_c$ being the ultraviolet momentum cut-off along the $k_z$-direction. We have chosen the momentum cutoff along the $k_z$ direction $k_c \sim \pi/a$ with $k_c > Q$ where $a$ is the

lattice constant. It is clear from the above equation that, like the real part, the imaginary part of $\sigma_{xx}$ also scales with $n$ in mWSMs. Since Eq. (10) is a complicated function of $\omega$, its exact behavior can be retrieved using numerics. The real and imaginary parts of $\sigma_{xx}$ from Eqs. (9) and (10) are plotted in Fig. 1(a), where the two frequencies $\omega_{1,2}$ are directly influenced by the tilt parameter $C$, which controls the region of vertical transitions. The amplitude of the vertical transitions here is directly proportional to the topological charge $n$, and it increases as one progresses from single WSMs to mWSMs, as can be seen directly from Eq. 9. It is worth noting that, while the low-energy model of the mWSM can accurately capture the real part of the conductivity for $\omega_c >> \omega$ (for example, Re[$\sigma_{xx}$], which is cut-off independent), the imaginary part of the conductivity becomes cut-off dependent that can be avoided by using a lattice regularization [42].

From Eq. (8), the Re[$\sigma_{zz}(\omega)$] is evaluated as

$$\text{Re}[\sigma_{zz}(\omega)] = \begin{cases} 0, & \text{for } 0 < \omega < \omega_1, \\ \mathcal{G}_{zz}\left[\frac{\sqrt{\pi}\,\Gamma(1+\frac{1}{n})}{2\Gamma(\frac{3}{2}+\frac{1}{n})} - x_{-2}F^1(1/2, -1/n, 3/2, x_-^2)\right], & \text{for } \omega_1 < \omega < \omega_2, \\ \mathcal{G}_{zz}\left[\frac{\sqrt{\pi}\,\Gamma(1+\frac{1}{n})}{\Gamma(\frac{3}{2}+\frac{1}{n})}\right], & \text{for } \omega > \omega_2, \end{cases} \quad (11)$$

where $x_- = 2\mu/\hbar\omega|C| - 1/|C|$ and $_2F^1(a,b,c,z)$ is a hypergeometric function. Unlike Re[$\sigma_{xx}$], here it is difficult to derive the explicit and generalized $n$ dependence of the Re[$\sigma_{zz}$]. With the help of numerics, we plot the real part of the $\sigma_{zz}$ in Fig. 1(b), for $n = 1, 2, 3$. Interestingly, we find that unlike single WSM ($n = 1$) case, for double WSM ($n = 2$), the Re[$\sigma_{zz}$] becomes frequency independent in the region $\omega > \omega_2$, whereas for triple WSM, it decays with increasing $\omega$. In particular, there exists a power law dependence in $\omega$ in this region, given by Re[$\sigma_{zz}$] $\propto \omega^{2/n-1}$ [36, 50]. Furthermore, for WSMs with $n > 1$, Re[$\sigma_{zz}$] increases more rapidly in the region $\omega_1 < \omega < \omega_2$. It is worth noting that the chiral or achiral tilt configuration has no effect on the diagonal components of the conductivity tensor, as confirmed by Eqs. (9)-(11).

We emphasize that determining an exact analytical expression of the Im[$\sigma_{zz}$] is cumbersome. The numerical integration of the following equation would help determine the imaginary part of the conductivity

$$\text{Im}[\sigma_{zz}(\omega)] = -\frac{2\omega}{\pi}\int_0^{\omega_c}\frac{\text{Re}[\sigma_{zz}(\omega') - \sigma_{zz}(0)]d\omega'}{\omega'^2 - \omega^2}, \quad (12)$$

which we plot for $n = 1, 2, 3$ in Fig. 1(b) (dotted curves). The figure shows that the magnitude of Im[$\sigma_{zz}$] for a single WSM is a nearly linear decreasing function of $\omega$, with a slight change in curvature at two photon bound frequencies. For $n = 2$ and 3, the magnitude of Im[$\sigma_{zz}$] enhances compared to $n = 1$ case. In addition, the nature of Im[$\sigma_{zz}$] in mWSMs is nonlinear with $\omega$, which shows a dip within the region $\omega_1 < \omega < \omega_2$.

## 3.2 Off-diagonal components of optical conductivity

The off-diagonal or transverse components of the optical conductivity tensor are derived here. It is clear from the Dirac identity that in order to avoid the principal term $P$, one has to calculate first the imaginary part, which takes the form

$$\text{Im}[\sigma_{mn}] = \sum_{s=\pm 1} s\frac{1}{(2\pi)^3}\int_0^{2\pi}d\phi_k\int_0^\infty k_\perp dk_\perp\int_{-k_c}^{k_c}dk_z$$
$$\times\left\{\text{Re}[(\mathcal{P}^{-+})_m]\text{Im}[(\mathcal{P}^{-+})_n] - \text{Im}[(\mathcal{P}^{-+})_m]\text{Re}[(\mathcal{P}^{-+})_n]\right\}$$
$$\times\frac{f_k^{eq}}{2\hbar\sqrt{\alpha_n^2 k_\perp^{2n} + v^2 k_z^2}}\pi\delta(\omega - \omega_k). \quad (13)$$

Note that, unlike the diagonal case where we calculated the conductivity for one node and summed it up for two, here we treat them individually. Now, proceeding with the $\phi_k$ integration (which leads to $\text{Im}[\sigma_{yz}] = 0 = \text{Im}[\sigma_{xz}]$) and applying the above mentioned change of variables to evaluate $k_\perp$ integration, we finally end up with the expression of $\text{Im}[\sigma_{xy}]$ of a mWSM containing two Weyl nodes with chiral tilt (i.e., $C_+ = -C_-$) of the form

$$\text{Im}[\sigma_{xy}] = \frac{e^2 n \mu^2}{8\pi\hbar^3 \omega \nu} \left( \sum_{s=\pm 1} s \int_{-\hbar\omega/2\mu}^{\hbar\omega/2\mu} x\, dx \sum_{p=\pm 1} p\, \Theta\left[ 1 - sCx - p\frac{\hbar\omega}{2\mu} \right] \right), \qquad (14)$$

which, upon simplification, gives closed analytical expressions for three frequency regions of type-I mWSMs as

$$\text{Im}[\sigma_{xy}(\omega)] = \begin{cases} 0, & \text{for } 0 < \omega < \omega_1, \\ \mathcal{G}_{zz}\left[ \frac{\sqrt{\pi}\,\Gamma(1+\frac{1}{n})}{2\Gamma(\frac{3}{2}+\frac{1}{n})} - x_{-2}F^1(1/2, -1/n, 3/2, x_-^2) \right], & \text{for } \omega_1 < \omega < \omega_2, \\ \mathcal{G}_{zz}\left[ \frac{\sqrt{\pi}\,\Gamma(1+\frac{1}{n})}{\Gamma(\frac{3}{2}+\frac{1}{n})} \right], & \text{for } \omega > \omega_2, \end{cases} \quad (15)$$

where $\kappa_o = \frac{1}{|C|^2}\left( \frac{\mu^2}{2\hbar^2\omega^2} - \frac{\mu}{2\hbar\omega} + \frac{1}{8} \right) - \frac{1}{8}$. Clearly, the imaginary part results from real optical transitions that are asymmetrically Pauli-blocked, which only exists in the frequency interval $\omega_1 < \omega < \omega_2$. In addition, it scales linearly with $n$ within the two photon energy bound frequencies ($\omega_{1,2}$). In Fig. 2(a), we depict the frequency-dependent behavior of $\text{Im}[\sigma_{xy}]$ for two relative orientations of the Weyl nodes ($C = \pm 0.4$), exhibiting negative and positive values (mirror image with respect to frequency-axis), respectively. It is clear from the figure that the magnitude of $\text{Im}[\sigma_{xy}]$ enhances with $n$, as also evident from Eq. (15).

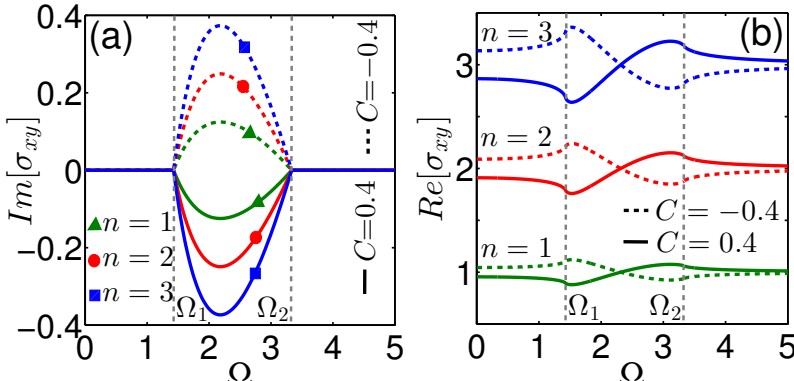

Figure 2: (a) Imaginary part of the $\sigma_{xy}$ as a function of rescaled frequency $\Omega$ of type-I mWSMs ($n = 1, 2, 3$) for $C = 0.4$ (solid cuves) and $C = -0.4$ (dashed curves). (b) Represents the same for real part of $\sigma_{xy}$. Here, the vertical dashed lines represent the two photon frequency bounds. The conductivities are rescaled by $e^2 Q/(\pi h)$, and the values of other parameters are the same as in Fig. 1. Here positive $C$ refers to the case: $C_s = s|C|$; and negative $C$ refers to the case: $C_s = -s|C|$.

We would like to point out that the real part of the Hall conductivity has two parts: i) ac or frequency dependent part ($\text{Re}[\sigma_{xy}^{\text{ac}}]$) and ii) dc or frequency independent part ($\text{Re}[\sigma_{xy}^{\text{dc}}]$), so that $\text{Re}[\sigma_{xy}] = \text{Re}[\sigma_{xy}^{\text{ac}}] + \text{Re}[\sigma_{xy}^{\text{dc}}]$ [32]. The real part of the conductivity is derived from

the Kramers-Kronig relation as

$$
\begin{aligned}
\mathrm{Re}[\sigma_{xy}] &= \frac{2}{\pi}\mathrm{P}\int_0^{\omega_c}\frac{\omega'\mathrm{Im}[\sigma_{xy}(\omega')]d\omega'}{(\omega'^2-\omega^2)} \\
&= \mathrm{sgn}(C)\sigma_\mu^n\left[\frac{1}{|C|}-\frac{1}{2|C|^2}\ln\left|\frac{(\omega_2^2-\omega^2)}{(\omega_1^2-\omega^2)}\right|\right. \\
&\left.+\left(\frac{\mu}{2\hbar\omega|C|^2}+\frac{\hbar\omega}{8\mu|C|^2}-\frac{\hbar\omega}{8\mu}\right)\ln\left|\frac{(\omega_2-\omega)(\omega_1+\omega)}{(\omega_2+\omega)(\omega_1-\omega)}\right|\right],
\end{aligned}
\tag{16}
$$

where $\sigma_\mu^n = e^2\mu n/(h^2 v)$.

It is important to note that, $\sigma_{xy}^{\mathrm{dc}}(\omega=0)$ (since the dc part is always real) has two contributions: i) intrinsic or 'universal' contribution $\sigma_{xy}^{(\mathrm{in})}$ and ii) free carrier contribution $\sigma_{xy}^{(\mathrm{free})}$. The $\sigma_{xy}^{(\mathrm{free})}$ can be extracted by taking the $\omega\to 0$ limit of the Eq. (16). On the other hand, the intrinsic part, arising from the separation between the Weyl nodes, can be written as $e^2 nQ/(\pi h)$ [51]. Therefore, the total dc contribution of $\sigma_{xy}$ is given by

$$
\begin{aligned}
\sigma_{xy}^{\mathrm{dc}}(\omega=0) &= \frac{e^2\mu n}{h^2 v}\left[\frac{2}{C}+\frac{1}{C^2}\ln\left(\frac{1-C}{1+C}\right)\right]+\frac{e^2 nQ}{\pi h} \\
&= \sigma_{xy}^{(\mathrm{free})}+\sigma_{xy}^{(\mathrm{in})}.
\end{aligned}
\tag{17}
$$

We would like to point out that the total $\sigma_{xy}^{\mathrm{dc}}(\omega=0)$ given in Eq. (17) can also be obtained by substituting $\omega=0$ in Eq. (3). Here, we have neglected the frequency dependent part of the intrinsic contribution. It is clear from the above expression that the $\sigma_{xy}^{(\mathrm{in})}$ is 'universal' in the sense that it only depends on the separation $Q$ between the opposite chirality Weyl nodes and is independent of $\mu$ and the tilt parameter $C$. The contribution $\sigma_{xy}^{(\mathrm{free})}$, on the other hand, comes from the free carriers present near the multi-Weyl nodes and is thus $\mu$-dependent. For mWSMs, these contributions are linearly proportional to $n$. In Fig. 2(b), we plot the total real part of the Hall conductivity $\mathrm{Re}[\sigma_{xy}]$ as a function of frequency for different $n$, which is positive across frequencies and possesses two discontinuities at two frequency bounds ($\omega_{1,2}$). Here, although $\mathrm{Re}[\sigma_{xy}^{\mathrm{ac}}]$ changes its sign, the total $\mathrm{Re}[\sigma_{xy}]$ remains positive due to the dc contribution.

# 4 Kerr and Faraday rotations from thin film of mWSMs

Since having calculated both the diagonal and off-diagonal conductivity tensors, we find a nonzero $ac$ Hall conductivity. Here, we investigate the Kerr and Faraday rotations of an ultrathin film of a type-I mWSM with a thickness ($d$) satisfying $a<<d<<\lambda$, where $\lambda$ is the wavelength of light. We now consider the case where the linearly polarized light incident on the mWSM surface and propagates along the separation between the Weyl nodes, i.e., along the $z$-direction as mentioned in Eq. (1), so that the polarization vector of the light lies on the $xy$ plane. Considering the interface of air and mWSM thin film at the $z=0$ plane, the components of the incident ($\mathbf{E_I}$), reflected ($\mathbf{E_R}$) and transmitted ($\mathbf{E_T}$) electric fields can be written as

$$
\begin{aligned}
\mathbf{E_I} &= (E_I^s, E_I^p\cos\theta_I, -E_I^p\sin\theta_I)e^{i(\mathbf{q_i}\cdot\mathbf{r}-\omega_i t)}, \\
\mathbf{E_R} &= (E_R^s, E_R^p\cos\theta_R, E_R^p\sin\theta_R)e^{i(\mathbf{q_r}\cdot\mathbf{r}-\omega_r t)}, \\
\mathbf{E_T} &= (E_T^s, E_T^p\cos\theta_T, -E_T^p\sin\theta_T)e^{i(\mathbf{q_t}\cdot\mathbf{r}-\omega_t t)},
\end{aligned}
\tag{18}
$$

where $\theta_I$, $\theta_R$, and $\theta_T$ are the angles, and $q_i$, $q_r$, and $q_t$ are the wave vectors of the incident, reflected, and transmitted electric fields, respectively. The superscripts $s$ and $p$ denote the s-polarization and p-polarization of the light, respectively. The corresponding magnetic fields can be obtained as $\mathbf{B} = (n_{\mathrm{ri}}/c)(\hat{\mathbf{k}} \times \mathbf{E})$, where $n_{\mathrm{ri}}$ is the refractive index of the medium. Considering that the medium on either side of the mWSM thin film is air, the electric and magnetic fields at the interface must satisfy the following boundary conditions:

$$(i)\ \mathbf{E}_1^{\|} - \mathbf{E}_2^{\|} = 0, \quad (ii)\ B_1^{\perp} - B_2^{\perp} = 0,$$
$$(iii)\ \mathbf{B}_1^{\|} - \mathbf{B}_2^{\|} = \mu_0 \mathbf{K}^{\mathrm{sh}} \times \hat{q}, \quad (iv)\ \epsilon_1 E_1^{\perp} - \epsilon_2 E_2^{\perp} = \sigma_f, \tag{19}$$

where $\hat{q}$ (i.e., $-\hat{z}$) is the unit vector normal to the interface pointing from medium-2 to medium-1, $\sigma_f$ is the free charge density, and $\mathbf{K}^{\mathrm{sh}}$ is the sheet or surface current. In this case, $\mathbf{E}_1$ and $\mathbf{E}_2$ ($\mathbf{B}_1$ and $\mathbf{B}_2$) represent the total electric (magnetic) fields in regions 1 and 2, respectively. In particular, $\mathbf{E}_1 = \mathbf{E}_I + \mathbf{E}_R$ and $\mathbf{E}_2 = \mathbf{E}_T$. The surface current density can be expressed as $K_i^{\mathrm{sh}} = \sigma_{ij}^d E_j$, where $\sigma_{ij}^d$ is the surface conductivity matrix. It is worth noting that the surface conductivity of the mWSM thin film is related to its bulk conductivity ($\sigma_{ij}$) via the expression $\sigma_{ij}^d = d\,\sigma_{ij}$ [42]. After solving the above-mentioned boundary conditions, the different reflection and transmission coefficients $\left[(r/t)_{ps,ss,sp,pp} = \left(\frac{E_{R/T}^{p,s,s,p}}{E_I^{s,s,p,p}}\right) \text{ at } E_I^{p,p,s,s} = 0\right]$ can be obtained as

$$r_{ps} = \frac{\cos\theta_T}{\cos\theta_I} t_{ps} = 2n_I\,\mathcal{S}\,\sigma_{yx}^d \cos\theta_I \cos\theta_T,$$
$$r_{ss} = t_{ss} - 1 = -2n_I\,\mathcal{S}\,\sigma_2^d \cos\theta_I - 1,$$
$$r_{sp} = t_{sp} = 2n_I\,\mathcal{S}\,\sigma_{xy}^d \cos\theta_I \cos\theta_T,$$
$$r_{pp} = \frac{\cos\theta_T}{\cos\theta_I} t_{pp} - 1 = -2n_I\,\mathcal{S}\,\sigma_1^d \cos\theta_T - 1, \tag{20}$$

where $\mathcal{S}^{-1} = c\mu_I(\sigma_{yx}^d \sigma_{xy}^d \cos\theta_I \cos\theta_T - \sigma_1^d \sigma_2^d)$ with $\sigma_1^d = \sigma_{xx}^d + n_I \cos\theta_I/(c\mu_I) + n_T \cos\theta_T/(c\mu_T)$, $\sigma_2^d = \sigma_{yy}^d \cos\theta_I \cos\theta_T + n_I \cos\theta_T/(c\mu_I) + n_T \cos\theta_I/(c\mu_T)$. Here, $n_I$ and $n_T$ are the refractive indices and $\mu_I$ and $\mu_T$ are the permeability of the incident and transmitted medium, respectively. It is to be noted that in our case, we find $\sigma_{xy}^d = -\sigma_{yx}^d$.

Using the above reflection and transmission coefficients, we can now calculate the Kerr and Faraday angles for the polarization rotation of the reflected and transmitted beams, respectively, due to the $s$ and $p$-polarized incident lights as [52]

$$\Phi_M^{s/p} = \frac{1}{2}\tan^{-1}\left(\frac{2Re[\chi_M^{s/p}]}{1 - |\chi_M^{s/p}|^2}\right),$$
$$\Psi_M^{s/p} = \frac{1}{2}\sin^{-1}\left(\frac{2Im[\chi_M^{s/p}]}{1 + |\chi_M^{s/p}|^2}\right), \tag{21}$$

where $M = K, F$ stand for Kerr and Faraday rotation and $\chi_M$ is a complex dimensionless quantity that can be expressed as: $\chi_K^s = \frac{r_{ps}}{r_{ss}}$, $\chi_K^p = -\frac{r_{sp}}{r_{pp}}$, $\chi_F^s = \frac{t_{ps}}{t_{ss}}$, and $\chi_F^p = -\frac{t_{sp}}{t_{pp}}$. The ellipticity angle $\Psi_M$ related to polarization rotation that measures the major-minor axis ratio of the polarization ellipse. The above expressions clearly show that the optical Hall conductivity ($\sigma_{xy}^d/\sigma_{yx}^d$) is solely responsible for the polarization rotation of both the reflected and transmitted light. This is due to the fact that the reflection coefficient $r_{sp}$ ($r_{ps}$) and transmission coefficient $t_{sp}$ ($t_{ps}$) are proportional to $\sigma_{xy}^d$ ($\sigma_{yx}^d$) and thus the $\chi_M$, which is $\propto r_{sp}/r_{ps}$ for Kerr rotation and $\propto t_{sp}/t_{ps}$ for Faraday rotation. As a result, the Kerr and Faraday rotations vanish in a TR-symmetric mWSM since the optical Hall conductivity is zero due to the presence of TRS. In addition, in the case of achiral-tilted mWSM (i.e., $C_+ = C_- = C$), the transverse

conductivity vanishes as can be seen from Eq. (14), resulting in the disappearance of Kerr and Faraday rotations. It is important to note that, due to the finite thickness $d$, two boundaries of the film can act as a Fabry-Perot cavity, where scattering from both the interfaces can lead to an interference effect, which in turn can modify Kerr rotation. The maxima and minima conditions for this interference effect are $d = l\lambda/2$ and $d = (2l+1)\lambda/4$, where $l$ is an integer. In this work, since $\lambda \gg d$, the phase difference is negligible, resulting in minimal impact on the Kerr rotation.

From Eq. (21), we find that the Kerr rotation and corresponding ellipticity in thin film mWSMs are independent of topological charge for any angle of incidence. This is because all of the reflection coefficients ($r$) are nearly proportional to $n$, and the $\chi_K$, which is defined by the ratio of $r$, becomes $n$-independent for $n = 1, 2, 3$. Interestingly, the polarization rotation of the transmitted light, i.e., the Faraday rotation and corresponding ellipticity angle, is found to be $n$-dependent. In particular, $\Phi_F$ enhance with increasing $n$. The reason for this is the following: the magnitude of $r_{ss}$ and $r_{pp}$ are very small i.e., $r_{ss}, r_{pp} << 1$, causing $t_{ss}$ and $t_{pp}$ to be independent of $n$ since they are related by $t_{ss} = 1 + r_{ss}$ and $t_{pp} = 1 + r_{pp}$, respectively. As a result, the $\chi_F$, which is defined by the ratio of $t$, becomes linearly proportional to $n$ for $n = 1, 2, 3$. Interestingly, the corresponding ellipticity scales with $n^2$ i.e., $\Psi_F^{mWSM} = n^2 \Psi_F^{n=1}$ in the regions $\omega < \omega_1$ and $\omega > \omega_2$, however it deviates from the $n^2$ scaling in the intermediate frequencies.

To get an estimate of $\Phi_K$ and $\Phi_F$, we consider normal incidence ($\theta_I = \theta_T = 0$), which forces $r_{ss} = r_{pp}$ as well as $t_{ss} = t_{pp}$. In this case, $\chi_K^{s/p}$ in thin mWSM ($d << \lambda$) is mainly determined by the ratio $\sigma_{xy}^d / \sigma_{xx}^d$. Since $\sigma_{xy}^d \propto nQ$ (lowest order in $Q$) and $\sigma_{xx}^d \propto n$ ($Q$ independent), $\chi_K^{s/p}$ is proportional to $Q$. Therefore, Kerr rotation is found to be very large ($\sim$ of the order of radians) for typical parameters of a WSM we use here. In contrast, $\Phi_K$ in ferromagnetic systems and topological insulators has been found to be of the order of microradians [53,54]. The Faraday angle, on the other hand, is of the order of a few degrees that is still very large compared to the case of topological insulators [53,54], where $\Phi_F$ has been found to be $< 1$ degree. We would like to point out that, the Kerr rotation is nearly independent of $d$ [41,42], to the contrary, in the case of Faraday rotation, the complex dimensionless quantity $\chi_F^{p/s}$ for normal incidence can be written as $\chi_F^{p/s} = \frac{\sigma_{xy}^d}{\sigma_1^d} \propto \frac{d\,\sigma_{xy}}{(1+\frac{d\,\sigma_{xx}}{2c\epsilon_0})} \sim d\,\sigma_{xy}$ for $d << \lambda$, implying that the Faraday rotation is proportional to $d$. In this work, we consider $d = 10 - 50$ nm and the wavelength of light $\lambda$ is varied till near infrared wavelength range which satisfies the criteria $a \ll d \ll \lambda$. For instance, we choose $\Omega(= \hbar\omega/\mu) = 5$ so that $\hbar\omega = 1$ eV and $\lambda \sim 1200$ nm. However, the polarization rotation angles can also be enhanced significantly by adjusting the parameters $Q$, and $\mu$ in mWSMs.

In Fig. 3(a), we plot $\Phi_K$ due to normal incidence as a function of $\omega$ and the corresponding ellipticity is depicted in Fig. 3(b) for three different tilt parameters. These plots reveal that, although $\Phi_K$ abruptly changes sign within the intermediate bound frequencies, $\Psi_K$ always remains negative. Furthermore, these two quantities are independent of $n$ and polarization states ($s/p$). The Faraday rotation and corresponding ellipticity are plotted in Figs. 3(c) and 3(d) respectively. Unlike the Kerr rotation, which disappears in the Pauli-blocked region, the Faraday rotation has a nonzero value that increases linearly with $n$ and always remains positive. In particular, it increases first in the partially Pauli-blocked region (intermediate frequency range), decreases in the upper region, and finally saturates for small $d$. It is clear from this figure that $\Phi_k$ is dominated by the Re$[\sigma_{xy}]$. Conversely, the ellipticity angle is oppressed by the Im$[\sigma_{xy}]$ and shows $n^2$ dependence in the lower and upper-frequency regions. The middle region, on the other hand, exhibits highly nonlinear behavior.

It is worth noting that when linearly polarized light incidents on the mWSM surface and propagates perpendicular to the separation between the Weyl nodes (i.e., $\perp$ to $z$-axis), the Kerr

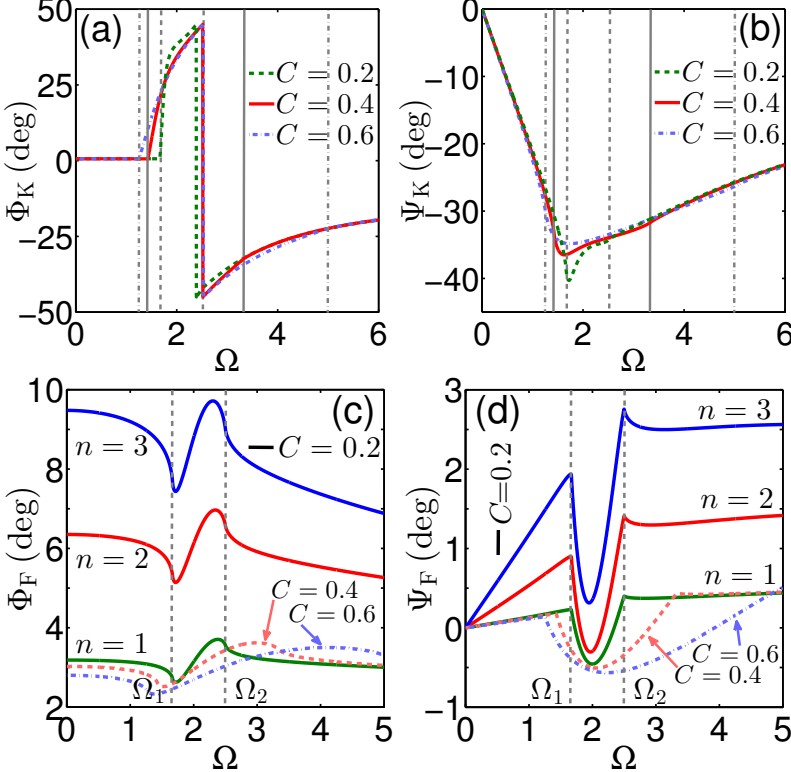

Figure 3: (a) Kerr angle and corresponding (b) ellipticity angle as a function of the rescaled optical frequency $\Omega$ for normal incidence for different strength of tilt parameters in thin film of mWSMs with $d = 10$nm and $\mu = 0.1eV$. (c) and (d) show the same for Faraday rotation with $d = 50$nm and $\mu = 0.2eV$. Here, all the other parameters are the same as in Fig. 1. Note that, in this case, $\Phi_K^p = \Phi_K^s = \Phi_K$ and $\Phi_F^p = \Phi_F^s = \Phi_F$. Interestingly, the figures show that although the Kerr rotation is $n$-independent, the Faraday angle increases linearly with $n$.

(Faraday) rotation vanishes because as the off-diagonal component of the conductivity tensor giving rise to reflection (transmission) coefficient becomes zero. This fact can help distinguish the surfaces of mWSMs with and without hosting Fermi arcs in experiments.

# 5 Kerr and Faraday rotations from the semi-infinite mWSMs

We now turn on the other limit $d \gg \lambda$ representing semi-infinite mWSM to investigate the polarization rotations. In this limit, Maxwell's equations get modified in the bulk due to $\mathbf{E}.\mathbf{B}$ coupling arising from topological properties [40,42,55]. This is a consequence of an additional axionic term in electromagnetic Lagrangian, $\delta L_\varphi = nc\epsilon_0\alpha_{\rm fs}\varphi\mathbf{E}.\mathbf{B}/\pi$, where $\alpha_{\rm fs}$ is the fine structure constant and $\varphi$ is the axionic field. Owing to the breaking of inversion and TRS, the axionic field $\varphi$ has a nontrivial space dependence, which is given by $\varphi = 2\mathbf{Q}.\mathbf{r} - 2Q_0 t$.

Here we restrict ourselves to the inversion-symmetric case, i.e., $Q_0 = 0$, and the axionic field $\varphi$ takes the integer multiple of $\pi$ due to its topological property. Using the modified electric and magnetic field in the bulk due to the new term in the Lagrangian, we obtain a

wave equation for a TRS broken mWSMs given by

$$\nabla \times (\nabla \times \mathbf{E}) = -\mu_p \sigma \frac{\partial \mathbf{E}}{\partial t} - \epsilon_p \mu_p \frac{\partial^2 \mathbf{E}}{\partial t^2} - \frac{2nc\epsilon_0 \mu_p \alpha_{\text{fs}}}{\pi} \mathbf{Q} \times \frac{\partial \mathbf{E}}{\partial t}, \qquad (22)$$

where $\mu_p$ is the permeability in the medium considered to be unchanged and $\epsilon_p = \epsilon_0 \epsilon_b$ is the permittivity, with $\epsilon_b$ representing the relative permittivity from the bound charge. The last term in Eq. (22), which is proportional to $Q$, is crucial for polarization rotation in bulk mWSMs. Here, depending on the light propagation direction, two distinct situations arise. In one case, the incident light propagates along the Fermi arc ($\hat{q} \parallel \mathbf{Q}$, say $\parallel \hat{z}$), i.e., onto the surface without Fermi arc. In the other case, the light is incident ($\hat{q} \perp \mathbf{Q}$, say $\parallel \hat{x}$) onto the surface containing Fermi arc. These two configurations correspond to Faraday and Voigt geometry, which we discuss in the following.

## 5.1 Faraday geometry

Let us consider the light incident on a surface without Fermi arcs, i.e., $\hat{q} \parallel Q\hat{z}$. The axionic bound charge density vanishes in this case because of $\mathbf{Q}.\mathbf{B} = 0$. Since $\mathbf{Q}$ behaves as the effective magnetization, this geometry corresponds to the Faraday configuration, which shows magneto-optic polar Kerr effects in magnetic systems. Starting with the incident electric field, $\mathbf{E_I} = E_0 e^{i(\mathbf{k}.\mathbf{r}-\omega t)}\hat{x}$, with $\mathbf{k} = (n_I \omega/c)\hat{z}$, the wave equation in the presence of the axion field [given in Eq. (22)] for the bulk mWSMs takes the following matrix form:

$$n_{\text{ri}}^2 \mathbf{E_1} = \begin{pmatrix} \epsilon'_{xx} & \epsilon'_{xy} & 0 \\ -\epsilon'_{xy} & \epsilon'_{yy} & 0 \\ 0 & 0 & \epsilon'_{zz} \end{pmatrix} \mathbf{E_2}, \qquad (23)$$

where $n_{\text{ri}}$ is the complex refractive index and $\mathbf{E_1} = (E_x, E_y, 0)$, $\mathbf{E_2} = (E_x, E_y, E_z)$. The dielectric tensor $\epsilon'_{ij}$, which is composed of optical conductivities and internode separation $Q$, can be written as

$$\epsilon'_{xx} = \epsilon'_{yy} = \epsilon_b + \frac{i}{\omega\epsilon_0}\sigma_{xx}, \qquad \epsilon'_{zz} = \epsilon_b + \frac{i}{\omega\epsilon_0}\sigma_{zz},$$

$$\epsilon'_{xy} = \frac{i}{\omega\epsilon_0}\sigma_{xy} + \frac{2in\alpha_{\text{fs}}c}{\pi}\frac{Q}{\omega} = \frac{i}{\omega\epsilon_0}(\sigma_{xy} + \sigma_{xy}^{(\text{in})}). \qquad (24)$$

The aforementioned equation [Eq. (24)] shows that, in contrast to diagonal components, the off-diagonal element $\text{Im}[\epsilon'_{xy}]$ contains the axionic contribution proportional to $Q$ together with the transverse conductivity. Additionally, the $\text{Im}[\epsilon'_{xy}]$ has a diverging feature for small $\omega$ regions ($\omega \to 0$), resulting in anomalous optical activities in the Pauli-blocked regime.

The solution of Eq. (23) leads to two eigenmodes as

$$n_+^2 = \epsilon'_{xx} + i\epsilon'_{xy}, \quad \text{and} \quad n_-^2 = \epsilon'_{xx} - i\epsilon'_{xy}, \qquad (25)$$

where $n_+$ and $n_-$ represent the refractive indices for the left and right circularly polarized eigenmodes in mWSMs, respectively. The electric fields corresponding to these transmitted modes can be expressed as $\mathbf{E_+} = t_+ E_0(\hat{x} + i\hat{y})e^{i\omega(t\mp n_+ z/c)}/\sqrt{2}$, $\mathbf{E_-} = t_- E_0(\hat{x} - i\hat{y})e^{i\omega(t\mp n_- z/c)}/\sqrt{2}$, where $t_+$ and $t_-$ are the transmission coefficients in the respective polarization directions. The reflected eigenmode in this configuration is given by $\mathbf{E_r} = E_0(r_x \hat{x} + ir_y \hat{y})e^{i\omega(t+n_I z/c)}$, where $r_x$ and $r_y$ are the reflection coefficients for the respective polarization directions. Using the boundary conditions given in Eqs. (19), we obtain [43]:

$$r_x = \frac{1 - n_- n_+}{1 + n_- + n_+ + n_- n_+},$$

$$r_y = \frac{i(n_- - n_+)}{1 + n_- + n_+ + n_- n_+}. \qquad (26)$$

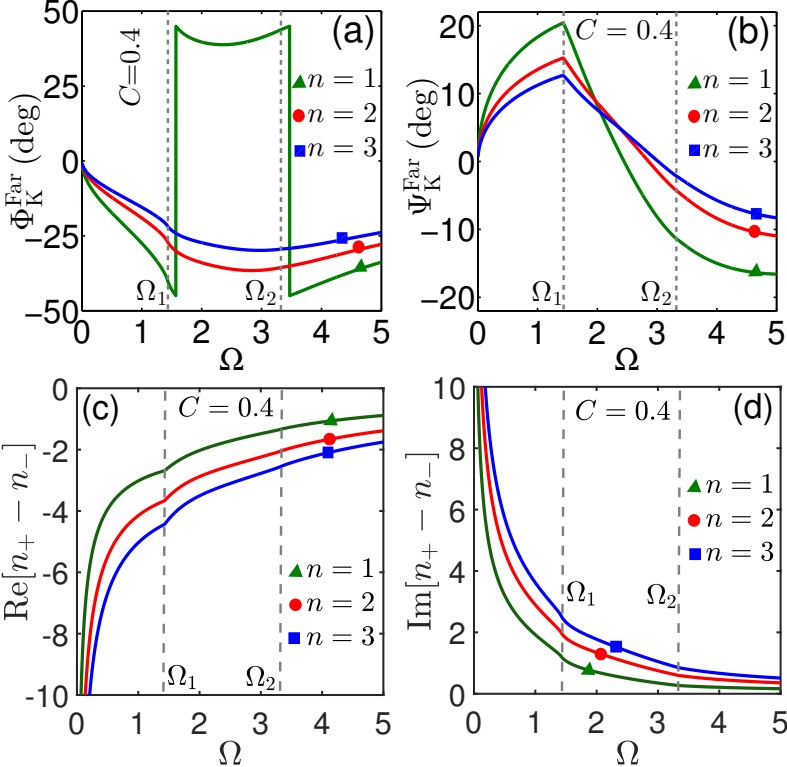

Figure 4: (a) Kerr angle and (b) corresponding ellipticity as a function of $\Omega$ in the Faraday configuration for a semi-infinite mWSM with $\mu = 0.1eV$ and $\epsilon_b = 1$. (c) and (d) show the variation of the real and imaginary parts of $(n_+ - n_-)$, which, respectively give rise to circular birefringence and circular dichroism. Here, all the other parameters are the same as in Fig. 1.

Defining the dimensionless quantity corresponding to Kerr ($\chi_K^F$) as

$$\chi_K^F = \frac{r_y}{r_x} = i\frac{n_+ - n_-}{n_+ n_- - 1} = i\frac{\sqrt{\epsilon'_{xx} + i\epsilon'_{xy}} - \sqrt{\epsilon'_{xx} - i\epsilon'_{xy}}}{\sqrt{\epsilon'_{xx} + i\epsilon'_{xy}}\sqrt{\epsilon'_{xx} - i\epsilon'_{xy}} - 1}, \tag{27}$$

one can obtain the Kerr angle from Eq. (21). It is clear from the Eq. (27) that, contrary to the case of thin film mWSMs, here, the polarization rotation depends on the axionic contribution as it appears in $\epsilon'_{xy}$. Interestingly, $\chi_K^F$ has linear-$Q$ dependence to the lowest order. Consequently, the polarization rotation manifests odd-$Q$ dependence, which is similar to the magneto-optic polar Kerr effect (odd in magnetization). In Fig. 4(a), we plot $\Phi_K^{\text{Far}}$ as a function of $\omega$ and the corresponding ellipticity $\Psi_K^{\text{Far}}$ is depicted in Fig. 4(b) for single, double and triple WSMs. Interestingly, in contrast to thin film case, $\Phi_K^{\text{Far}}$ and $\Psi_K^{\text{Far}}$ survive in the Pauli-blocked regime due to axionic contribution and their magnitude is suppressed with increasing $n$ for all frequency regimes as can be seen from Fig. 4.

In this geometry, the difference between the refractive indices of transmitted eigenmodes can give rise to circular birefringence and circular dichroism. Especially, $\text{Re}[n_+ - n_-]$ produces circular birefringence while $\text{Im}[n_+ - n_-]$ gives rise to circular dichroism in mWSMs. It is clear from Eq. (25) that both circular birefringence and circular dichroism increase with increasing $n$ compared to a single WSM. Interestingly, $n_+ - n_-$ is nonvanishing only for $\epsilon'_{xy} \neq 0$, and thus has the linear-$Q$ dependence to the lowest order. The $\text{Re}[n_+ - n_-]$ and $\text{Im}[n_+ - n_-]$ are depicted in Figs. 4(c) and 4(d), respectively. Since the circular birefringence and circular

dichroism are directly linked to the Faraday rotation angle and corresponding ellipticity by the following relations [56,57] $\phi_F^{Far} = \Phi_F^{Far} + i\Psi_F^{Far}$; $\Phi_F^{Far} = \frac{\pi d}{\lambda}Re[\Delta n]$ $\Psi_F^{Far} = \frac{\pi d}{\lambda}Im[\Delta n]$ ($\phi_F^{Far}$ is the complex Faraday rotation), one can easily verify our results on polarization rotation of transmitted light in an experiment by considering a bulk mWSM with thickness $d$ that meets the $d \gg \lambda$ criteria. Consequently, the Faraday rotation and ellipticity increase as $n$ increases.

It is worth mentioning that, although we have chosen $\epsilon_b = 1$ for the current work, it is a material-dependent parameter and can have higher values in WSMs, for example, $\epsilon_b = 6.2$ in TaAs [58]. However, for a large value of $\epsilon_b$, both $\chi_K^F$ and $\Delta n$ become very small in optical regime ($\chi_K^F, \Delta n \to 0$) because of $\epsilon'_{xx} \gg \epsilon'_{xy}$ as can be easily seen from Eq. (27), leading to tiny $\Phi_K^{Far}$ and $\Phi_F^{Far}$ respectively.

### 5.2 Voigt geometry

In this geometry, light is assumed to be incident on the surface containing Fermi arc states. Specifically, for our model, the light is incident on the $y-z$ plane and propagates along the $x$-direction, which is perpendicular to the Weyl-node separation $\mathbf{Q}$ (i.e., $Q\hat{z}$). To achieve non-zero polarization rotation, we choose the polarization of the incident light to be along $(\hat{y} + \hat{z})/\sqrt{2}$.

Now solving the light propagation equations in this geometry, we obtain two linearly polarized modes as

$$n_\perp^2 = \epsilon'_{yy} - \frac{\epsilon'^2_{xy}}{\epsilon'_{xx}}, \quad \text{and} \quad n_\parallel^2 = \epsilon'_{zz}, \tag{28}$$

where $n_\parallel$ and $n_\perp$ are the refractive indices of the two polarized modes propagating along and perpendicular to $\mathbf{Q}$ within the mWSMs. It is clear from Eq. (28) that the refractive index $n_\parallel^2$ is almost independent of $Q$, whereas $n_\perp^2$ is an even function of $Q$. The electric fields corresponding to these modes are given by $\mathbf{E}_\parallel = t_\parallel E_0 e^{i\omega(t \mp n_\parallel x/c)}\hat{z}$ and $\mathbf{E}_\perp = t_\perp E_0(\epsilon'_{xy}/\epsilon'_{xx}\hat{x} + \hat{y})e^{i\omega(t \mp n_\perp x/c)}$, respectively, where $E_0$ is the amplitude of the electric field, $t_\perp$ and $t_\parallel$ are the transmission coefficients in the respective polarization directions. Similarly, the electric field of the reflected mode can be expressed as $\mathbf{E}_r = E_0(r_\parallel \hat{z} + ir_\perp \hat{y})e^{i\omega(t + n_I x/c)}$, where $r_\perp$ and $r_\parallel$ are the reflection coefficients in the respective polarization directions.

Using the boundary conditions given in Eqs. (19), we finally obtain the reflection and transmission coefficients as

$$r_\parallel = \frac{1-n_\parallel}{\sqrt{2}(1+n_\parallel)}, \quad \text{and} \quad r_\perp = \frac{1-n_\perp}{\sqrt{2}(1+n_\perp)}. \tag{29}$$

Now defining the dimensionless quantities corresponding to the Kerr rotation as

$$\chi_K^V = \frac{r_\parallel}{r_\perp} = \frac{\left(1 - \sqrt{\epsilon'_{zz}}\right)\left(1 + \sqrt{\epsilon'_{yy} - \frac{\epsilon'^2_{xy}}{\epsilon'_{xx}}}\right)}{\left(1 + \sqrt{\epsilon'_{zz}}\right)\left(1 - \sqrt{\epsilon'_{yy} - \frac{\epsilon'^2_{xy}}{\epsilon'_{xx}}}\right)}, \tag{30}$$

the Kerr rotation angle, as well as corresponding ellipticity angle, can be obtained from Eq. (21). Although, similar to the Faraday geometry, $\chi_K^V$ here depends on the axionic contribution, however, it depicts quadratic-$Q$ dependence in lowest order which is in contrast to Faraday geometry. Consequently, the polarization rotation also becomes even function of $Q$, which is analogous to the Voigt effect. It is clear from the Eq. (28) that this configuration gives rise to linear birefringence and linear dichroism, defined as $Re[n_\parallel - n_\perp]$ and $Im[n_\parallel - n_\perp]$ respectively, increase with $n$ in mWSMs.

As the incident light is polarized by an initial angle $\Phi_0$ (here $\Phi_0 = \pi/4$), the measurement angle should be subtracted by $\Phi_0$. It is important to note that if the incident light is polarized

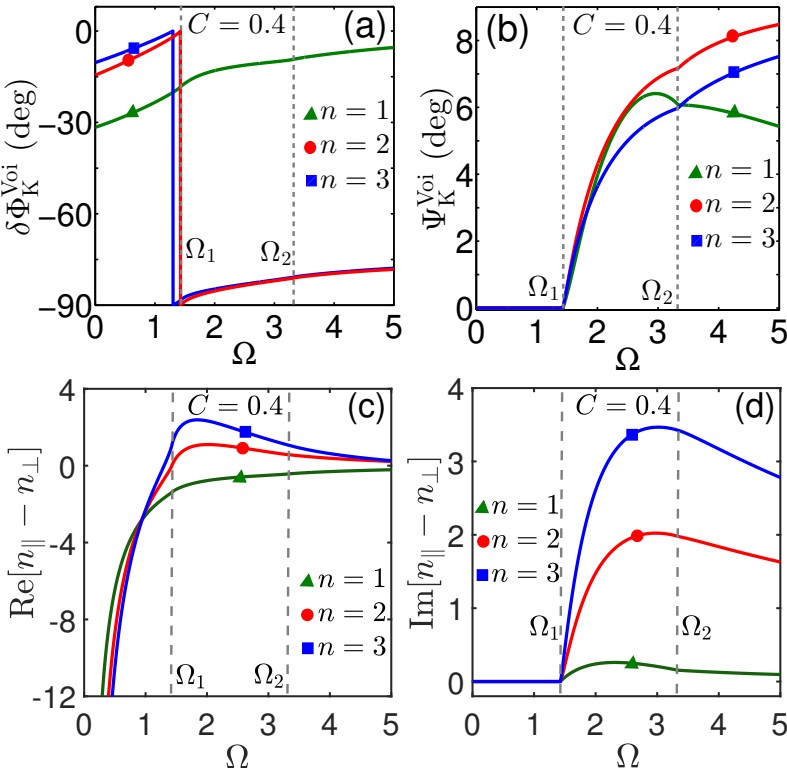

Figure 5: (a) Kerr angle and (b) corresponding ellipticity as a function of $\Omega$ in the Voigt configuration for a semi-infinite mWSM with $\mu = 0.1 eV$ and $\epsilon_b = 1$. (c) and (d) show the variation of the real and imaginary part of $(n_\parallel - n_\perp)$, which, respectively give rise to linear birefringence and linear dichroism. Here, $\Phi_0 = \pi/4$ and all the other parameters are the same as in Fig. 1.

along either the $\hat{y}$ or $\hat{z}$ directions, the boundary conditions will lead $r_\perp$ or $r_\parallel$ to become zero, resulting in zero polarization rotations. Hence, an incident polarization angle with the $\hat{y}$ and $\hat{z}$ axes is necessary.

In Figs. 5(a) and 5(b), we depict the Kerr angle $\delta\Phi_K^{\text{Voi}}$ ($= \Phi_K^{\text{Voi}} - \Phi_0$) and corresponding ellipticity $\Psi_K^{\text{Voi}}$ as a function of $\omega$ for single, double and triple WSMs. The real and imaginary parts of $(n_\parallel - n_\perp)$, which generate linear birefringence and linear dichroism, are shown in Figs. 5(c) and 5(d), respectively. Contrary to Faraday geometry, while $\delta\Phi_K^{\text{Voi}}$ and $\text{Re}[\Delta n]^{\text{Voi}}$ are finite in the Pauli-blocked regime, the ellipticity $\Psi_K^{\text{Voi}}$ and $\text{Im}[\Delta n]^{\text{Voi}}$ vanish. In the Pauli-blocked region, the magnitude of $\delta\Phi_K^{\text{Voi}}$ decreases, while $\text{Re}[\Delta n]^{\text{Voi}}$ increases as we go higher $n$. In the other two frequency regions, $\delta\Phi_K^{\text{Voi}}$ shows the opposite variation with $n$. The linear birefringence and linear dichroism lead to a polarization rotation (so called "Voigt rotation") of the transmitted light followed by the relation [59, 60]: $\phi_F^{\text{Voi}} \approx \frac{\pi d}{i\lambda}(n_\parallel - n_\perp)$, which can be measured in experiment for a bulk mWSM of thickness $d$. Moreover, these plots show the distinctive nature of mWSMs from a single WSM, which can be exploited to differentiate between them.

# 6 Discussion and Conclusion

We investigate the Kerr and Faraday rotations in type-I TRS broken mWSMs ($n > 1$) in the absence of an external magnetic field for two cases: (i) thin film limit ($a << d << \lambda$) and

(ii) semi-infinite limit ($d >> \lambda$), which we compare for a conventional WSM ($n = 1$). To put into perspective the fundamental qualitative difference between a conventional WSM and mWSM is that in a WSM, the dispersion around a Weyl node is isotropic in all momentum directions. While, for mWSMs, the dispersion around a double (triple) Weyl node becomes quadratic (cubic) along both $k_x$ and $k_y$ directions, with a linear variation along $k_z$. Using the low-energy model of mWSM, we analytically obtain the optical conductivity tensor including the impact of internode separation and finite $\mu$ within the framework of Kubo response theory. From our analytical calculations, we find that both the longitudinal and transverse components of the optical conductivity tensor are renormalized by the topological charge $n$. In particular, the longitudinal components ($\sigma_{xx}, \sigma_{yy}$) perpendicular to the Weyl node separation ($Q\hat{z}$) as well as the transverse component ($\sigma_{xy}$) are linearly proportional to $n$, while the component $\sigma_{zz}$, which is along the node separation, follows nontrivial dependence on $n$.

Using the obtained optical conductivity, in the case of thin film limit, we show that the Kerr rotation is mainly determined by the optical Hall conductivity and is proportional to the separation between the Weyl nodes of opposite chirality in a TRS broken mWSM. We also show that the polarization rotation of the reflected light is independent of topological charge $n$ and vanishes in Pauli-blocked region. In contrast, the Faraday rotation, oppressed by the Re[$\sigma_{xy}$], is finite in Pauli-blocked region and it depends on $n$. Remarkably, we find that $\Phi_F$ and corresponding ellipticity angle scale as $n$ and $n^2$ respectively in mWSM, which could possibly be employed to differentiate a conventional WSM from mWSM. In addition, we estimate the magnitude of both the Kerr and Faraday rotations turns out to be very large compared to other materials [53, 54].

On the other hand, in the case of semi-infinite mWSM, we explore the polarization rotation for two cases (1) Faraday geometry and (2) Voigt Geometry. Our analysis reveals that, in contrast to the thin film mWSMs, the polarization rotation in both cases is finite even in the Pauli-blocked regime and is dominated by the axion electrodynamics, which modifies Maxwell's equations. We show that polarization rotation is an odd function of $Q$ in Faraday geometry while showing even-$Q$ dependence in Voigt geometry. We further show that the magnitude of the Kerr angle in Faraday geometry decreases as $n$ increases. However, in the Voigt geometry, the magnitude of $\delta\Phi_K^{\text{Voi}}$ decreases with increasing $n$ in the Pauli-blocked region while in other two frequency regions, $\delta\Phi_K^{\text{Voi}}$ shows the opposite variation. In addition, the circular (linear) birefringence and circular (linear) dichroism in Faraday (Voigt) geometry enhance with the topological charge. Therefore, the polarization rotations could be used as a probe to distinguish single, double, and triple WSMs from each other in experiments. Furthermore, polarization rotation could help to discriminate the surfaces of mWSMs with and without hosting Fermi arcs. Note that, the trivial bands can also appear at the Fermi level in realistic Weyl materials. However, the presence of a trivial band at the Fermi level would not change the proposed results qualitatively since the transverse conductivity due to it is minimal compared to the nontrivial bands of mWSM in the absence of an external magnetic field. The magnetic WSM $Co_3Sn_2S_2$ [61,62], double WSM $HgCr_2Se_4$ [7,18] as well as cubic Dirac semimetal $A(MoX)_3$ [21] (with A = Rb, TI; X = Te) can be the possible candidates to show the proposed results on Kerr and Faraday rotations.

We would like to point out that contrary to the low-energy model used in this work, a real mWSM may consist of Weyl nodes with different tilts with respect to one another, and the number of pair of nodes can be greater than one. Besides, in the case of type-II mWSMs, the optical conductivity tensor calculated using the low-energy model becomes momentum cutoff-dependent. Since, it is well known that a lattice model of Weyl fermions with lattice regularization provides a natural ultraviolet cutoff to the low-energy Dirac spectrum, which is why, one needs to study a mWSM Hamiltonian using DFT or a lattice Hamiltonian to predict the quantitatively correct experimental behavior of polarization rotation in mWSMs. This is

an interesting question which we leave for future study. In addition, investigating the Kerr and Faraday rotations beyond Weyl systems [63, 64] such as triple component fermions, multi-fold fermions would also be a fascinating question to look into.

*Note added:* —Recently, we noticed one preprint [65] appeared in parallel with our work on Kerr rotation in multi-Weyl semimetals.

# Acknowledgements

The Authors acknowledge Bitan Roy (Lehigh University) and Andrea Marini (University of L'Aquila) for valuable discussions.

**Funding information**   The work at Los Alamos National Laboratory was carried out under the auspices of the US Department of Energy (DOE) National Nuclear Security Administration under Contract No. 89233218CNA000001. It was supported by the LANL LDRD Program, and in part by the Center for Integrated Nanotechnologies, a DOE BES user facility, in partnership with the LANL Institutional Computing Program for computational resources. S.G. and S.N. acknowledge the support by the National Science Foundation Grant No. DMR-2138008.

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
