# Peer review of "Theoretical investigations on Kerr and Faraday rotations in topological multi-Weyl Semimetals"

_SciPost Physics, doi:SciPost Phys. 15, 133 (2023)_

## Round 3 · Referee Report · Anonymous (Referee 1) · 2022-12-11

Strengths

  1. The paper provides a very detailed analysis of Kerr and Faraday rotation in mWSMs. It establishes a new electromagnetic response in mWSMs.
  2. It is very well written and sufficient details are provided for the analysis.

Weaknesses

  1. The paper doesn’t mention any possible candidate materials where this effect can be realized. Especially, the condition that there are only two-nodes with crage greater than one with finite tilt makes the model very restrictive.
  2. The paper doesn’t provide any analysis of energy scales. This is important because of many factors, especially the fact that continuum model is applicable only in a very tiny energy and momentum window for most Weyl semi-metals. Usually, there are many electronic bands in the vicinity of Wey lnode and it’s difficult to disentangle the contributions arising from trivial bands and topological band crossings. For most materials even a two-band tightbinding model fails to capture these essential features as even the matrix elements between topological trivial bands start to contribute to conductivity. It would be helpful if authors can provide some estimate of energy and momentum cutoffs they have used in the continuum model on the basis of electronic band structure of existing topological weyl semimemetals.

Report

This paper investigates the electromagnetic response of topological multi-Weyl semimetals. This paper satisfies the criteria of Scipost as this study opens up a new pathway to probe and utilize the topological features of multi-Weyl semimetals using electromagnetic field. Most importantly, authors explore the dependence of Kerr and Faraday rotation on the charge of the Weyl node.

I highly recommend this paper for publication in Scipost. However, there are certain issues that should be addressed: 1. I don’t understand how can one measure Faraday rotation in semi-infinite geometry. Could authors please clarify what is meant by Faraday angle in Eq. 28. The Kerr rotation can be measured directly from the quantities evaluated in Eq. 27 but Faraday rotation also depends on the path light traverses inside the material. Here, it’s not so clear how would one measure the effect on the other end of the sample in semi-infinite geometry. Maybe, one can consider a bulk sample but given that the system here exhibit circular dichroisma dn circular birfringence so the rotation would also depend on the length of the sample. If I’m not missing something important here, this issue seems very concerning. 2. Now, even, in the thin-film limit, the film thickness would play some role in deciding the Faraday rotation angle. The authors haved cited Ref. 51 which studies only Kerr and Faraday effect for monolayer graphene and hence the thickness of the sample doesn’t come into picture. It would be very helpful if authors can make some comments about the range of thickness and compare it to the wavelength of the light used which in turn would depend on the energy scales of the system which are not discussed here. Also, would Kerr rotation be modified in any way, if we also consider the reflection at the other end of the sample. 3.I would like to reemphasize that authors should also provide some estimate about the tilt as it decides the frequency range over which some of the quantities show a significantly important behavior. There is a minor typo in the paragraph below Eq. 9. The frequency range for the region in which the vertical transitions are Pauli unblocked should be $omega>omega_2$. 4.It would help if authors can provide the derivation for equation 16 and 17. It’s not so clear how they divide the real part of the off-diagonal conductivity into DC and AC part. It seems that equation 16 is non-zero even in the Dc limit. Another worrisome aspect of these equations (16-19) is that they diverge for zero tilt (C=0). Could authors please provide an interpretation for C=0 limit of these equations.

Overall the paper is very well written and the analysis is very informative.

  • validity: high
  • significance: good
  • originality: good
  • clarity: good
  • formatting: good
  • grammar: good

Author:  Snehasish Nandy  on 2023-02-12  [id 3346]

(in reply to Report 1 on 2022-12-11)

We are grateful to the Referee for his/her report, helpful comments and suggestions as well as positive assessment of our work. Please find our response to the comments from the Referee in the attached file.

Attachment:

Referee_Response.pdf

---

## Round 4 · Referee Report · Anonymous (Referee 2) · 2023-7-14

Report
The manuscript analyzes optical properties of a magnetic multi-Weyl semimetal and identifies the signatures of the order of the Weyl node (topological charge) by studying Faraday and Kerr effects.
In the first round of review another referee identified two limitations of the manuscript: the lack of discussion of possible materials and a missing discussion of energy scales as well as raised some technical issues. I have found that the authors have appropriately addressed these issues in the resubmitted version of the manuscript and in their report to the first referee.
I agree with the recommendation of the first referee to recommend the paper to SciPost Physics. However I believe the discussion in the manuscript would be made stronger if the authors explicitly explain:
- What are the qualitative differences between $n=1$ (conventional WSM) and $n>1$ (mWSM).
- Which of the signatures would remain visible in presence of additional trivial bands present on the Fermi level.
With these extensions I am happy to recommend the manuscript to SciPost Physics.

Author: Snehasish Nandy on 2023-07-21 [id 3826]
(in reply to Report 1 on 2023-07-14)We are grateful to the Referee for his/her report and recommending our work for publication. Please find our response to the comments from the Referee in the attached file.
Attachment:
Referee_Response_1.pdf

---

## Round 4 · Author Response

List of changes

---

## Round 4 · List of Changes

List of changes
Summary of changes in response to referee report
1) In response to referee comment 1, we have added a comment on possible candidate materials along with two new references (Refs. ~61 and 62) in the revised version of the manuscript. (First paragraph, Left column, Page 11)
2) In response to referee comment 2, we have added a comment on momentum cutoff of our low-energy model in the revised manuscript. (First Paragraph, Right Column, Page 4)
3) In response to referee comment 3, we have added clarifications on the polarization rotation of the transmitted light in both Faraday geometry and Voigt geometry. We have also discussed the experimental realization of polarization rotations using our results, and added new references (Refs.~56, 57, 59 and 60) in the revised manuscript (Second paragraph, Left column, Page 9; Third paragraph, Left column, Page 10). We have also modified the abstract and conclusions accordingly.
4) In response to referee comment 4, we have added clarifications on the role of thickness of the thin film and multiple reflection in Kerr rotation in the revised manuscript (First Paragraph, Left Column, Page 7). We have also added a comment on the range of thickness and wavelength of the incident light in the revised manuscript. (Third Paragraph, Left Column, Page 7)
5) In response to referee comment 5, we have added a comment on the tilt parameter (First paragraph, Left column, Page 4). We have also corrected the typo in the revised manuscript.
6) We have elaborated the off-diagonal conductivity part in the revised manuscript in response to referee comment 6 and added a new reference (Ref.~51). (Right Column, Page 5)

---

## Round 5 · Author Response

We are grateful to the Referee for the careful reading as well as for taking his/her time to write a report on our paper. We thank the referee for recommending our paper for publication. We have uploaded our response to the referee comments in separate file.

---

## Round 5 · List of Changes

Summary of changes in response to referee report

1) In response to referee comment 1, we have added a relevant comment on qualitative differences between a conventional WSM and mWSM in the revised manuscript. (First and second paragraph of the discussion and conclusion section, Right column, Page 10)

2) In response to referee comment 2, we have added a relevant comment on trivial band contribution to polarization rotation in the third paragraph of the discussion and conclusion section of the revised manuscript. (Left column, Page 11)

You are currently on this page

Resubmission 2209.11217v5 on 21 July 2023

---

## Editorial Decision

published